# Associations between Pharmacotherapy for Cardiovascular Diseases and Periodontitis

**DOI:** 10.3390/ijerph18020770

**Published:** 2021-01-18

**Authors:** Ewa Pająk-Łysek, Maciej Polak, Grzegorz Kopeć, Mateusz Podolec, Moïse Desvarieux, Andrzej Pająk, Joanna Zarzecka

**Affiliations:** 1Department of Conservative Dentistry with Endodontics, Institute of Dentistry, Jagiellonian University Medical College, 31-155 Kraków, Poland; j.zarzecka@uj.edu.pl; 2Department of Epidemiology and Population Studies, Institute of Public Health, Faculty of Health Sciences, Jagiellonian University Medical College, 31-066 Kraków, Poland; maciej.1.polak@uj.edu.pl (M.P.); andrzej.pajak@uj.edu.pl (A.P.); 3Pulmonary Circulation Centre, Department of Cardiac and Vascular Diseases, Faculty of Medicine, Jagiellonian University Medical College, 31-008 Kraków, Poland; g.kopec@uj.edu.pl; 4John Paul II Hospital, 31-202 Kraków, Poland; mateusz.podolec@uj.edu.pl; 5Department of Coronary Artery Disease and Heart Failure, Faculty of Medicine, Jagiellonian University Medical College, John Paul II Hospital, 31-008 Kraków, Poland; 6Department of Epidemiology, Mailman School of Public Health, Columbia University, New York, NY 10032, USA; md108@cumc.columbia.edu; 7INSERM UMR 1153, Center de Recherche Epidemiologie et Statistique Paris Sorbonne Cité (CRESS), METHODS Core, 75004 Paris, France

**Keywords:** periodontitis, cardioprotective drugs, angiotensin II receptor blockers, statins, acetylsalicylic acid

## Abstract

The goal of the study was to assess the relationship between cardioprotective medications, i.e., beta-blockers, angiotensin-converting enzyme inhibitors (ACEIs), calcium channel blockers (CCBs), angiotensin II receptor blockers (ARBs), statins, acetylsalicylic acid (ASA), and periodontitis (PD). Background: Xerostomia increases the risk of PD and is a side effect of some pharmacotherapies. Information about the effect of cardioprotective treatment of periodontal status is scarce. Methods: We studied 562 dentate residents of Krakow at the age of 50 to 70 years. Information about treatment was collected using a standardized questionnaire. The pocket depth and clinical attachment level (CAL) were used to ascertain PD. Multivariate logistic regression was applied to assess the relation between cardioprotective medications and PD. Results: PD was found in 74% of participants. The range of cardioprotective drug use among participants was 7% (ARBs) to 32% (beta-blockers). After adjusting for age, sex, number of teeth, smoking, and education, ASA’s use was related to a lower prevalence of PD in all dentate participants (odds ratio (OR) = 0.63, 95% confidence interval (CI): 0.40–0.99). The use of ARBs and statins was found to be associated with a higher prevalence of PD in persons having ≥6 teeth (odds ratio (OR) = 3.57, 95% CI: 1.06–11.99 and OR = 1.81, 95% CI: 1.03–3.16, respectively). Further adjustment for CVD risk factors, history of coronary heart disease, and other chronic diseases did not attenuate the results. There was no significant relation between PD and the use of other cardioprotective drugs.

## 1. Introduction

Periodontitis (PD) is a chronic inflammatory condition that arises from bacterial factors inhabiting dental biofilm and calculus, resulting in damage to teeth and surrounding soft tissues [1]. PD is responsible for tooth loss, reduced chewing function, and impaired esthetics. According to statements by the joint European Federation of Periodontology and the American Academy of Periodontology, PD’s prevalence increases with age. It is more prevalent in certain groups, i.e., smokers, patients with diabetes, the obese, and socially disadvantaged persons. Furthermore, PD contributes to social inequality, reduces the quality of life, and decreases overall health [2]. In Europe, it is estimated that PD affects more than 50% of the population [3]. Some studies suggest that PD is more frequent in Poland, affecting about 60% of the age group 35–44 years and about 70% of the age group 65–74 years [4,5].

The relationship between PD and cardiovascular disease (CVD) has been widely studied over the past few years [6,7,8]. There is evidence of an association between PD and myocardial infarction and stroke [9], and between PD and hypertension in patients with CVD [10,11]. The mechanism of the association between PD and CVD is not fully understood. It is known that PD affects the inflammatory host response and increases inflammatory markers (C-reactive protein), which decrease after effective treatment [12,13,14]. There is evidence that endothelial dysfunction, which plays a key role in developing arteriosclerotic plaques, is related to PD [15]. Some studies suggest an invasion of endothelial cells by periodontal microbes [16,17]. Indeed, Desvarieux et al. confirmed that an improvement in clinical and microbial periodontal status is associated with inhibited progression of intima–media thickness, which is a measure of early atherosclerosis [18,19].

It is also well-known that xerostomia increases PD’s risk as saliva’s amount and composition are crucial first lines of defense against pathogens, with disturbances favoring oral infections such as caries and PD [20]. Although studies suggest that salivary gland dysfunction should not be considered a normal aging process [21,22,23], xerostomia may be more prevalent in older age groups due to more prevalent multiple chronic diseases and subsequently larger consumption of multiple medications [22,23,24]. Some cardioprotective drugs, such as antihypertensive agents, cause xerostomia [25,26,27,28]. CVD patients, mostly aged over 50 years, regularly take medications such as beta-blockers, angiotensin-converting enzyme inhibitors (ACEIs), calcium channel blockers (CCBs), angiotensin II receptor blockers (ARBs), statins, and acetylsalicylic acid (ASA).

We hypothesized that prolonged cardioprotective medication use could negatively affect periodontal status. The goal of the present study was to assess the relationship between the use of cardioprotective medications, including beta-blockers, ACEIs, CCBs, ARBs, statins, ASA, and PD.

## 2. Materials and Methods

The study was conducted in accordance with the World Medical Association Declaration of Helsinki. All participants signed an informed consent. The survey protocol was approved by the Bioethics Committee of the Jagiellonian University.

This study was performed on a random subsample of 1008 persons (478 men and 530 women), between 50 and 75 years of age, from the 10,728 Polish participants of the Health Alcohol and Psychosocial Factors in Eastern Europe (HAPIEE) Project. Details of the methodology used in the HAPIEE study, which examined a representative sample of residents of Kraków, the second-largest city in Poland, have been published earlier [29,30]. A brief overview of the present study is given below.

Baseline information included data on health, lifestyle, diet (food frequency), socioeconomic circumstances and psychosocial factors. A short examination included measurement of anthropometric parameters, blood pressure, lung function, assessment of cognitive function, and collection of a fasting venous blood sample. Re-examination included collection of data on healthy aging, economic wellbeing, medical history, and current medication status by face-to-face interview [29]. Additionally, in a subsample, oral examination was performed. Oral examination was carried out at the University Dental Clinic in Kraków, according to WHO recommendations [31], by dentists carefully trained in the study methods, and the results of the measurements were subjected to quality control. Standard extraoral and intraoral examinations were undertaken. The intraoral examination was carried out from the first to the fourth quadrant. Data were collected on the number of teeth with caries, extracted teeth, filled teeth, and use of partial and full dentures.

Periodontal clinical parameters were measured for each tooth, except for the third molars. Probing was done on six tooth sites with a 0.25 N force using a periodontal Hu–Friedy probe (model PCPUNC157). Patients were classified according to the broadly used case definition of PD developed by the CDC Periodontal Disease Surveillance Workgroup, based on pocket depth measurements and clinical attachment loss (CAL). CAL (distance from the cemento-enamel junction (CEJ) to the bottom of the pocket/sulcus) was determined using the distance from the CEJ to the free gingival margin (FGM) and the distance from the FGM to the bottom of the pocket/sulcus [31]. For the present report, participants were classified as having PD if criteria for moderate or severe PD were fulfilled, i.e., presence of ≥2 interproximal sites with CAL ≥ 4 mm (not on the same tooth) or ≥2 interproximal sites with pocket depth ≥ 5 mm (not on the same tooth) and presence of ≥2 interproximal sites with CAL ≥ 6 mm (not on the same tooth) and ≥2 interproximal sites with pocket depth ≥ 5 mm (not on the same tooth), respectively [32,33].

Information on age, education (primary, secondary, and higher), smoking status (current, former, or never), history of chronic diseases, and current use of cardioprotective medication was collected using the standardized questionnaire [29]. To assess the current use of cardioprotective medication, participants were asked to present documentation of their treatment or boxes of the drugs consumed. Consumed drugs were classified into the appropriate groups by a medical doctor.

Height and weight were measured according to the standard protocol. Participants with a body mass index (BMI) ≥ 30 kg/m^2^ were considered obese.

After at least 5 min of rest, blood pressure was measured in a sitting position and on the right arm using an Omron M5-1 device. Three measurements in 2 min intervals were taken. The mean of the second and third measurements was used in the analysis. Participants were classified as having hypertension if their blood pressure was ≥140/90 mmHg [19,34] or were taking blood pressure lowering medication within two weeks prior to the examination. Glucose concentration was determined in venous blood collected from participants fasting for at least 12 h. Participants were classified as having diabetes if their fasting plasma glucose was ≥7.0 mmol/L [35] or diabetes was diagnosed by their doctor. Blood lipids were determined using automated enzymatic methods. Low-density lipoprotein cholesterol (LDL-C) concentration was calculated using the Freidewald formula. Hypercholesterolemia was defined as LDL-C ≥ 3 mmol/L or total cholesterol ≥ 5 mmol/L [19,36,37].

Continuous variables were presented as means (standard deviation). The Shapiro–Wilk test was used to assess conformity with a normal distribution. Categorical variables were reported as absolute numbers and percentages, and the differences were tested using the chi-squared test. Differences in mean age and number of teeth were compared with the *t*-test. Multivariate logistic regression was applied to determine which medications were significantly related to poor periodontal status (PD ≥ 6 and CAL ≥ 6 mm/periodontitis). The analysis was carried out in all dentate (at least one tooth) persons and repeated after restriction to persons who had at least six teeth (residual dentition) to meet the WHO recommendations for dental clinical examination and to follow the practice of the other studies [31,38]. Statistical analyses were conducted using SPSS Statistics for Windows, version 25.0, IBM CORP. released 2017, Armonk, NY, USA. Statistical significance was accepted at the level *p* < 0.05.

## 3. Results

### 3.1. Descriptive Characteristics According to Dentition Levels

Of the 1008 persons invited to participate, 909 (430 men and 479 women) underwent an oral examination. Excluded from the further analysis were 347 edentulous participants, among whom none had implants. The analysis was carried out in 562 dentate (one tooth at least) persons (268 men and 294 women). The analysis was then repeated after restriction to 514 persons (251 men and 263 women) who had at least six teeth (residual dentition). Sample selection recruitment and restriction are presented in Appendix A. In dentate participants, the average age was 63 (SD = 6.3) years. The mean number of teeth was 16 (SD = 6.9). There were 38.0% of participants with a university education, and 21.4% were current smokers. The cardioprotective medication was used by 64% of participants. Nearly half of them were using two or three drugs, and over 11% were using more than three cardioprotective drugs. Periodontitis was found in 74.4%. In the subsample after restriction to persons with at least six teeth, the average age, mean number of teeth, frequency of use cardioprotective drugs, and distribution of covariates were similar to all dentate participants. In the group after restriction, the mean number of teeth was slightly greater at 17 (SD = 5.9) than in all dentate participants, which was 16 (SD = 6.9), but the prevalence of PD was similar, 76.7% and 74.4%, respectively. (Table 1).

The frequency of use of particular cardioprotective drugs was similar before and after restriction. Beta-blockers were the most frequently used (32.9% in dentate individuals and 31.3% in individuals with residual teeth) and ARBs were the least—about 7% in both groups (Table 2).

There was no significant difference in the proportion of participants with PD between participants with positive and negative history chronic diseases, which are the main indications for using cardioprotective drugs, neither in all dentate persons nor in persons with ≥6 teeth present. For a few participants, information on the history of chronic diseases was missing, so the analysis that used these data involved slightly fewer participants (Table 3).

### 3.2. Cardioprotective Medication Use and PD

The percentage of participants with PD using cardioprotective medications is presented in Table 4.

In the multivariate analyses adjusting for age and sex, all dentate participants using ARBs had greater than two and half times higher prevalence of PD (odds ratio (OR) = 2.66, 95% confidence interval (95%CI): 1.01–7.00); the relation was even stronger among persons with at least six teeth (OR = 3.63, 95% CI: 1.08–12.20). After further controlling for education, smoking, number of teeth, and hypertension or heart failure, the relation between ARBs use and PD remained significant among participants with at least six teeth (OR = 3.32, 95% CI: 1.01–11.27 and OR = 3.38, 95% CI: 1.001–11.38). In all dentate participants, the estimate of effect remained strong but was no longer statistically significant. Similarly, participants with at least six teeth who were using statins had a higher prevalence of PD by over 80% after controlling for age, sex, education, smoking, number of teeth. Further controlling for hypercholesterolemia, history of myocardial infarction, or his-tory of hospitalized coronary heart disease did not change the results materially (OR = 1.80, 95% CI: 1.03–3.15; OR = 1.80, 95% CI: 1.01–3.17; and OR = 1.73, 95% CI: 1.01–3.04). While not achieving statistical significance, the direction and estimates of OR were similar in all dentate persons (Table 5).

Finally, there was an inverse relationship between the use of ASA and PD (OR = 0.59, 95% CI: 0.38–0.92) in all dentate participants. Adjustment for age, sex, education, smoking, number of teeth, and history of myocardial infarction or history of hospitalized coronary heart or hospitalized chronic osteoarthritis did not attenuate the relation. The direction and estimate of effect were similar in participants with at least six teeth but not achieving statistical significance (Table 5).

Adjustment for the history of other chronic diseases and obesity did not change results materially for ARBs, statins, and ASA. There was no significant relation between PD and the use of remaining drugs studied; however, beta-blockers, ACEIs and CCBs, but not diuretics, had estimates of effect tending to increase PD prevalence.

## 4. Discussions

In this study, we found a positive association between the use of cardioprotective medication, specifically ARBs and statins, and PD prevalence. The relationships were stronger with larger dentition. Conversely, the use of ASA was inversely related to the prevalence of PD. These relationships were independent of CVD risk factors and a history of chronic diseases, which usually indicate cardioprotective drugs.

ARBs that block type 1 angiotensin II receptors on blood vessels and cardiac muscle are commonly used to treat hypertension and are considered safe, with no important side effects. There is no evidence that poor periodontal status is related to ARB use, to the best of our knowledge; however, our findings are in line with some previous observations. Several studies have shown a consistent increase in the prevalence of periodontitis in patients with hypertension or metabolic syndrome [39,40], patients treated for CVD [41], and patients using antipsychotic medications, which is related to slow salivary rate [42]. The reduced salivary flow might be a key factor in the relationship between the use of ARBs and PD. Xerostomia is a well-known side effect of ARBs use, and it is well-known that poor periodontal status is related to salivary hypofunction and xerostomia [43]. The lack of antibodies present in saliva may contribute to the development of periodontal disease caused by bacteria such as *Porphyromonas gingivalis*, a key pathogen occurring in the polymicrobial biofilm associated with chronic periodontitis [44]. Some studies have shown that ARBs are associated with decreased inflammation markers [45,46,47]; however, the association between ARBs and inflammatory markers was not confirmed in another study [48].

The results on statins, however, might be surprising. Statins were found to decrease biomarkers of periodontal disease [49]. Some evidence exists that adjunct statin therapy improves the effects of periodontal treatment [50,51,52]; however, a part of this evidence is based on animal studies with experimentally induced PD. Studies on humans mostly used local statin (gel) and had a short follow-up time. Few studies attempted to assess the effect of oral treatment with low or moderate doses of statins. However, they were done in small groups; the follow-up time did not exceed a couple of months, and the risk of bias due to confounding was not eliminated [52]. Such evidence does not seem to be convincing for longitudinal statin treatment. Earlier, the retrospective cohort study results suggesting that statin therapy prevented tooth loss were not confirmed after adjustment for confounders [53,54]. Our results are in accordance with a large Finnish cross-sectional study in which statin medication was associated with an increased likelihood of having teeth with deepened periodontal pockets among subjects with no gingival bleeding, and to a lesser extent among subjects with no dental plaque. Although in the same study, a weak negative association between statin medication and periodontal infection among subjects with a dental plaque or gingival bleeding was found [55]. In contrast to ARBs, statins are considered to positively influence inflammation markers and have an anti-inflammatory effect [49]. It is possible that the anti-inflammatory effect of statins could explain their weaker relation with PD compared to ARBs.

ASA is a known nonsteroid anti-inflammatory drug that acts through reducing the prostaglandins, and the negative relationship between its use and PD has been known for a long time [56]. Our results are in accordance with this knowledge.

It is also worth noting that, even while not achieving statistical significance, all of the drugs studied had estimates of effect tending to increase PD, with the exception of diuretics and ASA. This consistency in direction seems to warrant further studies in larger and various samples.

Our study, due to its cross-sectional design, does not allow for any firm conclusion on causality. There are other limitations in the interpretation of our results. First, our analysis excluded 347 edentulous participants (with no implants), which was more than one-third of all participants who underwent a dental examination. Although in Poland, tooth loss is mostly due to caries, some excluded patients may have lost teeth due to advanced PD; therefore, the relationships between the use of cardioprotective medications and PD could be even stronger than reported. Further, periodontal indicators depend on the number of teeth and are more reliable if calculated in individuals with at least six teeth [31]. We addressed this problem by repeating the analysis after restriction to persons having six teeth at least and by adjusting for the number of teeth in the multivariate analysis. Second, evaluation of PD was strictly based on measures of clinical parameters due to the difficulty in acquiring all parameters necessary for more detailed classification introduced at the 2017 World Workshop on the Classification of Periodontal and Peri-Implant Diseases and Conditions [57]. Third, our study did not allow us to address the problems of the individual dosage, frequency of intake, duration of use of particular drugs, and a time that had elapsed on the diagnoses of CVD and medication, which would require either a different study design or larger sample size to assure statistical power.

Despite the limitations mentioned above, the study has important strengths. It targeted a random sample of the inhabitants of the second-largest city in Poland, at the age at which most CVD incidents cased occur in Poland [58]. We used standardized research methods, which allowed for controlling important possible confounders (the results were stable in various statistical models using different combinations of covariates). To our knowledge, we are the first to assess the relationship between cardiovascular treatment and PD in the general population in Poland. Furthermore, it contributes to the interdisciplinary understanding of the relations between periodontal disease and the consequences of chronic, systemic diseases in the population at age over 50.

Statins are the most important drugs in the treatment of hypercholesterolemia. As the alternative for ACE inhibitors, ARBs are often recommended as the first-choice drugs in treating hypertension [19,34]. Both groups of drugs play a key role in preventing cardiovascular disease, and the lifesaving effect outweights all adverse side effects. It is unlikely that finding the positive relationship between their use and PD can affect the current recommendation for hypercholesterolemia and hypertension treatment. Nevertheless, if our findings are confirmed at other sites, a call for more intensive monitoring of periodontal status and PD prevention in patients treated with statins and ARBs would be reasonable. Considerations to medication choice in the treatment of hypertension in persons with PD might also be concerned.

Finally, PD is considered a risk factor for CVD [19]. However, the extensive prospective studies that found the relation between PD and incidence of CVD did not control the use of statins, ARBs, or other cardioprotective agents [59,60]. Our results suggest that the relation between PD and incidence of CVD might be explained partially by using cardioprotective drugs that cause xerostomia.

## 5. Conclusions

The use of ARBs and statins was found to be related to a higher prevalence of PD. The findings seem plausible but will need to be replicated in various settings, ideally with prospective studies to confirm the association. The inverse relation between the prevalence of PD and ASA’s use supports the postulated protective effect on periodontal tissues.

## Figures and Tables

**Table 1 ijerph-18-00770-t001:** Characteristics of the dentate population of the cohort.

	All Dentate (≥1 Tooth)		≥6 Teeth Present	
Mean age [years] (SD)	63	(6.3)	62	(6.3)
Gender				
Men [%]	268	47.7	251	48.8
Women [%]	294	52.3	263	51.2
Education [n, %]				
Primary	133	23.8	114	22.2
Secondary	214	38.2	193	37.6
University	213	38.0	206	40.1
Smokers				
Current [n, %]	120	21.4	108	21.1
Former [n, %]	150	26.8	137	26.8
Never [n, %]	290	51.8	267	52.1
Obesity (BMI ≥ 30) [n, %]	171	30.4	154	30.0
Diabetes [n, %]	80	14.2	67	13.0
Hypertension [n, %]	332	59.3	299	58.4
Hypercholesterolemia [n, %]	459	81.7	416	80.9
Myocardial infarction [n, %]	27	4.8	23	4.55
Hospitalized coronary heart disease [n, %]	42	7.5	37	7.3
Hospitalized heart failure [n, %]	8	1.4	6	1.2
Stroke [n, %]	7	1.3	5	1.3
Hospitalized chronic osteoarthritis [n, %]	32	5.7	30	5.9
Number of cardioprotective medications				
none [n, %]	202	35.9	191	37.2
1 [n, %]	123	21.9	113	22.0
2–3 [n, %]	170	30.2	150	29.2
>3 [n, %]	67	11.9	60	11.7
Mean number of teeth (SD)	16	(6.9)	17	(5.9)
PD [n, %]	418	74.4	394	76.7

BMI = body mass index, PD = periodontitis.

**Table 2 ijerph-18-00770-t002:** Distribution of cardioprotective medications use across the dentate population.

Cardioprotective Medications	All Dentate (≥1 Tooth)	≥6 Teeth Present
Beta-blockers [%]	32.9	31.3
ACEIs [%]	29.2	28.8
CCBs [%]	12.5	12.5
ARBs [%]	6.9	6.8
Diuretics [%]	18.9	19.1
Statins [%]	22.8	21.8
ASA [%]	22.8	21.0

ACEIs = angiotensin-converting enzyme inhibitors, CCBs = calcium channel blockers, ARBs = angiotensin II receptor blockers, ASA = acetylsalicylic acid.

**Table 3 ijerph-18-00770-t003:** Percentage of participants with periodontitis (% PD) by history of chronic diseases in all dentate persons and persons with ≥6 teeth present.

History of the Disease		All Dentate (≥1 Tooth)			>6 Teeth Present		
		N	% PD	*p*	N	% PD	*p*
Myocardial infarction	no	533	74.1	0.67	489	76.30	0.48
	yes	27	77.8		23	82.60	
Hospitalized coronary heart disease	no	516	74.2	0.78	473	76.30	0.51
	yes	42	76.2		37	81.10	
Hospitalized heart failure	no	554	74.2	0.65	508	76.40	0.34
	yes	8	87.5		6	100.00	
Stroke	no	551	74.2	0.99	505	76.40	0.99
	yes	7	71.4		5	80.00	
Hospitalized chronic osteoarthritis	no	527	74.6	0.47	481	76.90	0.39
	yes	32	68.8		30	70.00	
Diabetes	no	482	74.1	0.68	447	76.30	0.61
	yes	80	76.3		67	79.10	
Obesity	no	391	73.7	0.55	360	75.80	0.50
	yes	171	76.0		154	78.60	
Hypertension	no	228	71.5	0.21	213	72.30	0.06
	yes	332	76.2		299	79.60	
Hypercholesterolemia	no	103	76.7	0.55	98	77.60	0.82
	yes	459	73.9		416	76.40	

**Table 4 ijerph-18-00770-t004:** Percentage of participants with periodontitis (% PD) using cardioprotective medications.

		All Dentate (≥1 Tooth)			≥6 Teeth Present		
		[n]	[%]	*p*	[n]	[%]	*p*
Beta bloker	Not used	277	73.5	0.48	263	74.5	0.09
	Used	141	76.2		131	81.4	
ACEIs	Not used	293	73.6		276	75.4	
	Used	125	76.2	0.52	118	79.7	0.29
CCBs	Not used	363	73.8		344	76.4	
	Used	55	78.6	0.39	50	78.1	0.77
ARBs	Not used	384	73.4		362	75.6	
	Used	34	87.2	0.06	32	91.4	0.03
Diuretics	Not used	339	74.3		317	76.2	
	Used	79	74.5	0.97	77	78.6	0.62
Statins	Not used	315	72.6		300	74.6	
	Used	103	80.5	0.07	94	83.9	0.04
ASA	Not used	332	76.5		317	78.1	
	Used	86	67.2	0.03	77	71.3	0.14

*p* < 0.05. ACEIs = angiotensin-converting enzyme inhibitors, CCBs = calcium channel blockers. ARBs = angiotensin II receptor blockers, ASA = acetylsalicylic acid.

**Table 5 ijerph-18-00770-t005:** Relationship between use of cardioprotective medications and prevalence of periodontitis (PD) (reference group: nonusers or users of other drugs).

Cardioprotective Medication		All Dentate	≥6 Teeth Present
		(≥1 Tooth)	
		OR (95% CI)	OR (95% CI)
Beta-blockers	Model A	1.19 (0.78–1.80)	1.49 (0.93–2.39)
	Model B	1.25 (0.82–1.90)	1.50 (0.93–2.40)
ACEIs	Model A	1.09 (0.70–1.67)	1.19 (0.74–1.92)
	Model B	1.14 (0.73–1.76)	1.21 (0.75–1.96)
CCBs	Model A	1.24 (0.67–2.28)	0.99 (0.54–1.95)
	Model B	1.30 (0.70–2.43)	1.03 (0.54–1.96)
ARBs	Model A	2.66 (1.01–7.00) *	3.63 (1.08–12.20) *
	Model B	2.53 (0.96–6.71)	3.57 (1.06–11.99) *
	Model B + hypertension	2.42 (0.90–6.48)	3.32 (1.01–11.27) *
	Model B + history of heart failure	2.35 (0.88–6.27)	3.38 (1.001–11.38) *
Diuretics	Model A	0.98 (0.60–1.60)	1.10 (0.65–1.89)
	Model B	1.00 (0.61–1.65)	1.12 (0.65–1.93)
Statins	Model A	1.54 (0.95–2.52)	1.74 (1.00–3.05) *
	Model B	1.64 (0.99–2.70)	1.81 (1.03–3.16) *
	Model B + history of myocardial infarction	1.64 (0.99–2.71)	1.80 (1.01–3.17) *
	Model B + hypercholesterolemia	1.64 (0.99–2.69)	1.80 (1.03–3.15) *
	Model B + of hospitalized coronary heart disease	1.59 (0.96–2.63)	1.73 (1.01–3.04) *
ASA	Model A	0.59 (0.38–0.92) *	0.63 (0.38–1.03)
	Model B	0.63 (0.40–0.99) *	0.64 (0.39–1.05)
	Model B + history of myocardial infarction	0.64 (0.42–0.999) *	0.64 (0.39–1.05)
	Model B + history of hospitalized coronary heart disease	0.63 (0.40–0.998) *	0.62 (0.38–1.02)
	Model B + history of hospitalized chronic osteoathritis	0.64 (0.41–0.999) *	0.65 (0.39–1.07)

* *p* < 0.05. ACEIs = angiotensin-converting enzyme inhibitors. CCBs = calcium channel blockers. ARBs = angiotensin II receptor blockers. ASA = acetylsalicylic acid. OR = odds ratio; CI = confidence interval. Model A: adjusted for age and sex; Model B: adjusted for age, sex, education, smoking, and number of teeth.

## Data Availability

Data that support the findings of this study are available from the corresponding author upon reasonable request due to ethical and privacy restrictions.

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
