# Peer review of "Associations between Pharmacotherapy for Cardiovascular Diseases and Periodontitis"

_ijerph, 2021, doi:10.3390/ijerph18020770_

Round 1

Reviewer 1 Report

I described some comments as major revision for this article.

Dear authors, thank you for this interesting MS that the associations between pharmacotherapy and periodontitis. Please note the following points that need your attention:

Introduction: P2, Line 68: You described about xerostomia. Why did you measure the saliva flow?

Material & Methods: P2, Line 83: You picked up persons between 50 and 75 years old. Why did you select this age group? You should show the reference.

P3, Line 112, 115, 118: You classified the hypertension, diabetes and hypercholesterolemia. Are these criteria often used by medical authority and medical institution in your country? Please show the reference of these criteria.

You should describe how collect the information of participant’s medication. Did you take a questionnaire from participants? Please show in detail.

Results: Please show Flowchart for clarity of participant’s information.

P3, Line 135-137: You analyzed the 562 all dentate group and the 514 at least 6teeth present group. This is selection bias. Can you show the reference? You should explain why select this subgroup (at least 6teeth present group).

Table: Please add the number (n=562, n=514) of all dentate and 6 teeth present at the title column.

Table 1: Please add the row of “hypertension" and "hypercholesterolemia”.

Table 3: I think that the total number should be 562 or 514. But myocardial infarction, hospitalized coronary heart disease, stroke hospitalized chronic osteoarthritis and hypertension are not so. Is this mistype or data missing?

Table 3 and 4: Please add the (*) to indicate significance.

Sincerely,

Author Response

Dear Reviewer,

We do appreciate your comments, which allowed us to re-think our concepts and improve our paper. Below, please find our answers to your comments and indications to changes in the manuscript which were introduced.

Introduction: P2, Line 68: You described about xerostomia. Why did you measure the saliva flow?

Regrettably the saliva flow was not measured. Actually, it was not a part of the study hypothesis. However, the relations between some cardioprotective drugs and xerostomia and between xerostomia and periodontal disease were described well in the other studies (citations 20-28). This suggests that xerostomia might mediate the relation between cardioprotective treatment and periodontal disease.  We introduced our study and debated the results in the context of this knowledge.

Material & Methods: P2, Line 83: You picked up persons between 50 and 75 years old. Why did you select this age group? You should show the reference.

The age span covers 5 of 5-years age groups and refers to upper 10 years of middle age and first 15 years of old age. In Poland the peak number of incident cases and hospitalizations due CHD was found in the middle age group i.e. 60-64 years. We added a reference [ref. 58 ] and a comment was added in the discussion section (lines 263-267).

P3, Line 112, 115, 118: You classified the hypertension, diabetes and hypercholesterolemia. Are these criteria often used by medical authority and medical institution in your country? Please show the reference of these criteria.

According to the resent and earlier ESC/ESH guidelines, hypertension was defined as office SBP values >140 mmHg and/or diastolic BP (DBP) values >90 mmHg and BP below 140/90 mmHg is considered a primary treatment target for all hypertensive patients [ref.: 34, 19]. We are aware that last ACC/AHA/AAPA/ABC/ACPM/AGS/APhA/ASH/ASPC/NMA/PCNA Guideline define stage 1 hypertension for the values 130-139 and 80-89 mmHg but we assumed that this definition had no impact for the diagnosis and treatment of hypertension in Poland at the time of the survey. It seems reasonable and it was introduced in the earlier studies to classify as hypertensives those who report than had been informed about having high blood pressure by the doctor and had been taking blood pressure lowering treatment.

The cut-off point of fasting glucose > 7 mmol/l for diabetes is broadly accepted after being recommended by WHO [35]. Again it is reasonable and introduced in the other studies to classify as diabetics those  who reported that had diabetes diagnosed by a doctor and had been treated with anti-diabetic agents.  

Cut-off points LDL-C>3 mmol/l and TC>5 mmol/l were introduced by ESC and 9 other international scientific societies. Recent  versions of recommendations stress on the need to relate the diagnostic and treatment target levels with global CVD risk. However, LDL-C analysis remains the recommended as the primary lipid analysis method for screening, diagnosis and management, and cut-off point of 3 mmol/l is accepted for use in low risk group. TC is still recommended to be used for the estimation of total CV risk by means of the SCORE system and SCORE tables include a value of 5 mmol/l as one of the main cut-off points. References [Perk J 2012, Piepoli M  2016, Mach 2020]. LDL-C was not determined directly in our study but calculated using the Fredewald formula which is not applicable in case of major hypertriglyceridemia. It is likely that eliminating such dyslipidemic cases would bias the results more than classifying them on the basis of total cholesterol. 

Trying to answer the question whether these criteria are often used by medical authority and medical institution in Poland? I would tell that all the recommendations mentioned above were either translated into Polish and/or most important statements are reflected in the guidelines of Polish scientific societies.

We added all the citations mentioned above and to the current version of a paper.

You should describe how collect the information of participant’s medication. Did you take a questionnaire from participants? Please show in detail.

Information was collected by interview according to the standard questionnaire. To facilitate the interview, participants were asked to bring a documentation of their treatments or the boxes of the drugs consumed. The drugs were classified into the appropriate group by the medical doctor. The information was added to the methods section (lines 111-113).

Results: Please show Flowchart for clarity of participant’s information.

Added as supplementary figure 1.

P3, Line 135-137: You analyzed the 562 all dentate group and the 514 at least 6teeth present group. This is selection bias. Can you show the reference? You should explain why select this subgroup (at least 6teeth present group).

Edentulous persons had to be excluded as they are not at risk of periodontal disease  by definition.  The first analysis was done for all dentate participants i.e. having 1 tooth at least and then the analysis was repeated with a restriction to persons having 6 teeth at least (residual dentition).  Such restriction followed recommendations for oral examination for periodontal disease [ref. 31].

Table 3: Please add the number (n=562, n=514) of all dentate and 6 teeth present at the title column

The numbers 562 and 514 refer to all dentate participants and participants  with 6 teeth at least respectively. For the diseases. there were some missing data as some participants refused to report (quite often in the observational studied). The numbers “n=562” and “n=514” were added in the current version. Additionally, we added clarification to the numbers in the columns in the table 3, in the results section (lines 41-42).

Table 1: Please add the row of “hypertension" and "hypercholesterolemia”.

Added in the current version (table 1).

Table 3: I think that the total number should be 562 or 514. But myocardial infarction, hospitalized coronary heart disease, stroke hospitalized chronic osteoarthritis and hypertension are not so. Is this mistype or data missing?

The numbers 562 and 514 refer to all dentate participants and participants  with 6 teeth at least respectively. For the history of the diseases. there were few missing data as some participants refused to report (quite often in the observational studied). We added a comment in the result section (lines 65-67).

Table 3 and 4: Please add the (*) to indicate significance.

For table 3 all the differences were not significant which is indicated by the exact p value in the appropriate columns. For table 4. Three significant p values are marked with (*). The explanatory footnote was added below the table.

Reviewer 2 Report

Dear Authors,

I have major concerns regarding the methods of the study. Although the authors relates the current article with the published articles (ref no. 29 and 30), it is not clear if the informations for the current article was obtained during the re-examination period between 2006-2008. No supplementary on the questionnaires used for ref no 30 and the current article so I am not sure whether they repeated the questionnaire and clinical assessment again or used the same data as ref no. 30. The materials & methods section is difficult to understand in terms of classifications of each variables assessed in this study. The authors can use table to simplify the classsification and improve readers' understandings. Sentences 128-131 should have been mentioned at the beginning of the Materials & Methods paragraph.

Results: I would prefer if they use the same Table format as presented in ref.no 30 because in the current article, some informations are missing in Table 1 e.g. gender: only Men was reported, similarly with the education level and smoking status. 

Conclusion: Since they presumed the effects of the medications on periodontal disease through xerostomia, they should have measure the salivary flow rates too to confirm this associations. 

Author Response

Dear Reviewer,

Thank you for your comments, which allowed us to re-think our concepts and introduce introduce changes to the manuscript if feasible. Below, please find our answers.

I have major concerns regarding the methods of the study. Although the authors relates the current article with the published articles (ref no. 29 and 30), it is not clear if the informations for the current article was obtained during the re-examination period between 2006-2008. No supplementary on the questionnaires used for ref no 30 and the current article so I am not sure whether they repeated the questionnaire and clinical assessment again or used the same data as ref no. 30. The materials & methods section is difficult to understand in terms of classifications of each variables assessed in this study. The authors can use table to simplify the classsification and improve readers' understandings.

Indeed, oral examination and interview was done at Wave 2. Details for the collection of other information are given in ref. 29. Following to the above comment, we added a supplementary figure 1. Which should clarify the sampling, recruitment and restrictions to the sample. Additional clarification was added into the section on methods (lines 91-97).

Sentences 128-131 should have been mentioned at the beginning of the Materials & Methods paragraph.

The text “The study was conducted in accordance with the World Medical Association Declaration of Helsinki. All participants signed an informed consent. The survey protocol was approved by the Bioethics Committee of the Jagiellonian University. “ was reordered and moved to the beginning of the section (lines 77-79).

Results: I would prefer if they use the same Table format as presented in ref.no 30 because in the current article, some informations are missing in Table 1 e.g. gender: only Men was reported, similarly with the education level and smoking status.

We agree that the descriptive statistics by gender might be interesting. However, they were published earlier partially (ref 30). In the previous report the analysis which tested the main hypothesis was done separately by sex. In the present paper splitting the group by sex was not done and sex was used as one of covariates in the regression analysis (due to small numbers in the cells). Moreover, there was no significant interaction between medication and gender (gender does not moderate the relationship between medication and PD). For the above reasons, we would be in favor to leave it as in the first version of the manuscript and would appreciate the acceptance. However, in a consequence of the review procedure Table 1 was supplemented with new data.

Conclusion: Since they presumed the effects of the medications on periodontal disease through xerostomia, they should have measure the salivary flow rates too to confirm this associations.

We aimed to find out the relation between the use of cardioprotective medication  and periodontal disease. Xerostomia itself was not a part of the study hypothesis. However, we discussed our results in the context of the strong evidence from the other studies on the relations between cardioprotective drugs and xerostomia and between xerostomia and periodontal disease (ref. 42-43) which indicates the possible role of xerostomia as a mediator. As we did not have data on salivary flow, mentioning of “xerostomia” was deleted from the final conclusions.

Reviewer 3 Report

Overall, interesting article but requires important adjustments to add to its merit.

Introduction:

  • Reference 6 has been misused. Perhaps the authors meant to use reference 7? Also, there are more recent references you can use here. Please update references in general. 
  • How did you come up with the hypertension BP cutoff point being 140/90?

Materials and Methods:

-Please list you inclusion and exclusion criteria clearly.

-include more info on the diagnoses acquired on systemic condition. For instance, frequency of CVD, individual characteristics as feasible.

-Were the edentulous areas restored with implants at all? Please add a statement to confirm if implants are there or not. If so, these also need to be addressed (at least in the discussion).

Results:

  • I like table 1 summary. In regards to obesity, was waist to hip ratio record
  • Please revisit the percentages. There are multiple errors. The % should add up to a 100. Include number of non-diabetic for instance as well. 
  • For p values, substitute the , for .
  • CRPs were not mentioned here. The reader is assumed to have that recorded. Please provide them. This will add merit to your study (and address this in the discussion as CRP levels have been incorporated in the 2017 world workshop grading system). Although not assessed here, it should be referred to.
  • Doses and frequencies of medications.
  • Please state clearly that PD evaluation was strictly based on clinical parameters (as you also mentioned via CDC). Please refer to the 2017 world workshop definitions and state that due to the difficulty in acquiring all parameters, these case definitions were not used, rather the CDC/AAP last case definitions published were used.

Discussion:

-Include updated references. 

-Doses and frequencies were mentioned here, I believe it is important to mention that in the results. 

-Upon inclusion, what time had elapsed on the diagnoses of CVD and medication. Time factor unto the point of assessment should be included. 

-One point to emphasize is the interdisciplinary understanding and work required between periodontal and systemic health. 

Author Response

Dear Reviewer,

We do appreciate your comments, which allowed us to re-think our concepts and introduce important changes to the manuscript. Below, please find our answers.

Introduction:

  • Reference 6 has been misused. Perhaps the authors meant to use reference 7? Also, there are more recent references you can use here. Please update references in general.

We thank you very much for pointing out the mistake. The numbers of references were corrected In the current version.

  • How did you come up with the hypertension BP cutoff point being 140/90?

According to the resent and earlier ESC/ESH guidelines [ref. 19, 34], hypertension was defined as office SBP values >140 mmHg and/or diastolic BP (DBP) values >90 mmHg, and BP below 140/90 mmHg is considered a primary treatment target for all hypertensive patients .

Materials and Methods:

-Please list you inclusion and exclusion criteria clearly.

We carried out a population based, observational  study in residents of Krakow town. Besides being a resident of Krakow, at specific age range and ageing for participation there was no other inclusion criteria. In the course of the study there were two restrictions made. The first, the sample was restricted to dentate participants (excluded edentulous). The second, was the exclusion of participants having less than 6 teeth (for a part of the analysis).

For clarification of the selection and recruitment process we added a flow-chart as a supplementary figure 1.  We hope that current description supplemented by flow-chart provides all the information needed.

-include more info on the diagnoses acquired on systemic condition. For instance, frequency of CVD, individual characteristics as feasible.

The frequencies of the diseases were added to the table 1. History of the diseases was based on the interview according to the standard questionnaire.

-Were the edentulous areas restored with implants at all? Please add a statement to confirm if implants are there or not. If so, these also need to be addressed (at least in the discussion).

In fact, edentulous participants had no implants. This did not surprise us, as implants are expensive and not refunded in Poland. Further, as documented in the reference 30, “no teeth” status was related to low SES. Information on lack of implants was added in the results section (lines 141-142).

Results:

  • I like table 1 summary. In regards to obesity, was waist to hip ratio record

Regrettably, we are not able to provide data on waist to hip ratio. However, BMI was used in the multivariable analysis.

  • Please revisit the percentages. There are multiple errors. The % should add up to a 100. Include number of non-diabetic for instance as well. 

We checked all percentages in the table 1. They add to 100% where appropriate (small difference in decimals appear due to rounding).

In table 2. There were many participants who used more than 1 drug. Percent of users of particular drugs are independent of each other.

In table 3. Percent of participants with periodontal disease  in those who had particular diseases is  independent of the percent of PD in remaining persons who did not have the disease. They do not add to 100%.

In table 4 percent of participants withe periodontal disease  in those who use particular drugs is independent of the percent of PD in non-users. They do not add to 100%.

  • For p values, substitute the , for .

Corrected in the current version.

  • CRPs were not mentioned here. The reader is assumed to have that recorded. Please provide them. This will add merit to your study (and address this in the discussion as CRP levels have been incorporated in the 2017 world workshop grading system). Although not assessed here, it should be referred to.

We followed the recommendation below and explained that PD evaluation was based on clinical parameters assessed by oral examination, The grading of periodontitis introduced at the 2017 World Workshop was not applied. Regrettably, we are not able to provide information on CRP.

  • Doses and frequencies of medications.

Regrettably such detailed information was not collected. A comment was added in the discussion section as a limitation of the study (lines 259-262).

  • Please state clearly that PD evaluation was strictly based on clinical parameters (as you also mentioned via CDC). Please refer to the 2017 world workshop definitions and state that due to the difficulty in acquiring all parameters, these case definitions were not used, rather the CDC/AAP last case definitions published were used.

We do appreciate the comment. Actually, the corresponding text was added in the discussion section (lines 256-259).

Discussion:

-Include updated references.

New references were added (ref. 57-60).

-Doses and frequencies were mentioned here, I believe it is important to mention that in the results.

A problem of individual dosage was mentioned as we referred to the findings of the other studies. In the current version of our paper not addressing the problem of individual dosage and frequency of intake was mentioned in the discussion as a study limitation (lines 259-262).

-Upon inclusion, what time had elapsed on the diagnoses of CVD and medication. Time factor unto the point of assessment should be included.

A comment was added in the discussion section as a study limitation (lines 260-262).

-One point to emphasize is the interdisciplinary understanding and work required between periodontal and systemic health.

Thank you very much for this comment. We added the appropriate text in the discussion (lines 269-271).

Round 2

Reviewer 1 Report

Thanks for your efforts.

Just a minor edit here.

1, Table 3: I was able to understand that there were few missing data on table 3. So that, the number "n=562" and "n=514" is not needed.

2, Table 5: Please add the "*" to indicate significance. Are there significances in all results?

Author Response

Dear Reviewer,

Thank you very much for your comments. Below please find our comments and changes in the manuscript.

1, Table 3: I was able to understand that there were few missing data on table 3. So that, the number "n=562" and "n=514" is not needed.

Numbers "n=562" and "n=514" have been removed from the table 3 (lines 169)

2, Table 5: Please add the "*" to indicate significance. Are there significances in all results?

The "*" has been added to table 5 and the symbol was explained at the bottom of the table 5 (lines 193)